# On the Wake Dynamics of an Oscillating Cylinder via Proper Orthogonal Decomposition

Benet Eiximeno [1], Arnau Miró [1], Juan Carlos Cajas [2], Oriol Lehmkuhl [1] and Ivette Rodriguez [3,*]

1 Barcelona Supercomputing Center, 08034 Barcelona, Spain
2 ENES-Unidad Mérida, Universidad Nacional Autónoma de México, Mérida 97118, Yucatán, Mexico
3 Turbulence and Aerodynamics Research Group, Universitat Politècnica de Catalunya (UPC), 08221 Barcelona, Spain
* Correspondence: ivette.rodriguez@upc.edu

**Abstract:** The coherent structures and wake dynamics of a two-degree-of-freedom vibrating cylinder with a low mass ratio at $Re = 5300$ are investigated by means of proper orthogonal decomposition (POD) of a numerical database generated using large-eddy simulations. Two different reduced velocities of $U^* = 3.0$ and $U^* = 5.5$, which correspond with the initial and super-upper branches, are considered. This is the first time that this kind of analysis is performed in this kind of system in order to understand the role of large coherent motions on the amplification of the forces. In both branches of response, almost 1000 non-correlated in-time velocity fields have been decomposed using the snapshot method. It is seen that a large number of modes is required to represent 95% of the turbulent kinetic energy of the flow, but the first two modes contain a large percentage of the energy as they represent the wake large-scale vortex tubes. The energy dispersion of the high-order modes is attributed to the cylinder movement in the inline and cross-stream directions. Substantially different POD modes have been found in the two branches. While the first six modes resemble those observed in the static cylinder or in the initial branch of a one-degree of freedom cylinder in the initial branch, the modes not only contain information about the wake vortexes in the super-upper branch but also about the formation of the 2T vortex pattern and the Taylor–Görtler structures. It is shown that the 2T vortex pattern is formed by the interplay between the Taylor–Görtler stream-wise vortical structures and the cylinder movement and is responsible for the increase in the lift force and larger elongation in the super-upper branch.

**Keywords:** LES; two-degrees-of-freedom vibrating cylinder; POD; coherent structures

## 1. Introduction

Vortex-induced vibration (VIV) is an interesting topic in several fields of engineering as the vibrations generated by a fluid flow can have serious effects on the performance and structural stability of marine and land vehicles, oil tubes in sea platforms, civil engineering structures, and any tethered structures in the ocean [1,2]. More recently, VIV has also been considered as a source of renewable energy harvesting and there is currently ongoing research to develop both low-power devices [3–5] and commercial high-power ones [6–8].

Many research papers and comprehensive reviews on VIV have been published in the literature [9–14]. One of the most studied geometries in VIV is a circular cylinder forced or free to oscillate either with 1 or 2 degrees of freedom (DoF), as it is the most general geometry in many structures, such as those mentioned above. In the systems with 1 DoF, the cylinder can only move in the cross-stream direction, while it can also oscillate along the stream-wise direction with 2 DoF.

The system has different branches of response depending on the number of degrees of freedom, the mass ratio of the cylinder ($m^* = 4m/(\rho \pi D^2 L)$, a parameter that relates the mass of the cylinder with the mass of an equivalent volume of fluid, and the reduced velocity ($U^* = U_\infty/(f_n D)$) (which is the inverse of the natural frequency non-dimensionalised

with the free-stream velocity and the cylinder diameter). In the present work, a 2 DoF circular cylinder with a low mass ratio at two different reduced velocities is considered.

In the particular case of a 2 DoF cylinder with a low mass ratio ($m^* < 6$), and depending on the reduced velocity, three branches of response are observed: (i) the initial branch, (ii) the super-upper branch, and (iii) the lower branch (see Figure 1). The first branch of response of the system, which occurs at low reduced velocities, is the initial branch (I). The cylinder has a low amplitude of oscillation in both stream-wise and cross-stream directions. The vortex shedding gives rise to a 2S wake pattern, following the Williamson and Roshko [15] classification of the wake patterns, i.e., in every shedding cycle, two single counter-rotating vortices are formed, similar to the von Kármán vortex street observed in the static cylinder (see Figure 1).

With an increase in the reduced velocity, the system enters in what is known as the super-upper (SU) branch. In the SU branch, the system reaches its higher oscillation amplitudes (up to 1.5$D$). Moreover, this higher amplitude of response has also been associated with the 2T wake mode, i.e., two triplets of vortices per shedding cycle, which maximises the energy transfer between the structure and the fluid [16,17]. The 2T and multi-vortex-shedding modes are also related to the appearance of a third and higher harmonics in the lift force [18]. Finally, the lower branch occurs after the super-upper branch and before the de-synchronisation. Its movement amplitudes are lower than those of the SU branch and it has a 2P wake mode, i.e., two pairs of counter-rotating vortices per cycle.

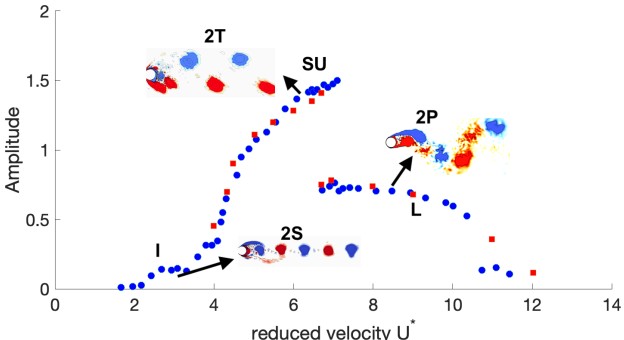

**Figure 1.** Wake topology depending on the reduced velocity for a 2 DoF cylinder with a low mass ratio. I—initial branch, SU—super-upper branch, and L—lower branch. 2S, 2T, and 2P are the wake topologies, as defined by Williamson and Roshko [15]. Data from (blue circles) Jauvtis and Williamson [17], (red squares) Pastrana et al. [16].

Jauvtis and Williamson [17] performed experimental tests with the aim of characterizing the flow dynamics of the 2 DoF oscillating cylinder with a low mass ratio. They were the first to identify the 2T wake pattern and characterize the super-upper branch. Later, Gsell et al. [19] performed direct numerical simulations at $Re_D = 3900$, paying special attention to the phasing and spectral content of the fluid forcing. More recently, Pastrana et al. [16] conducted large-eddy simulations at a range of Reynolds numbers between $Re_D = 3900$ and $Re_D = 11,000$ in order to obtain insight into the fluid dynamics and wake patterns for the three different branches of response. A later study performed by the same authors showed that when the cylinder is at its maximum elongation in the super-upper branch, Taylor–Görtler structures are formed on the cylinder surface. The appearance of these structures can be attributed to a centrifugal instability occurring twice every vortex-shedding cycle [20].

As already discussed, the VIV phenomenon generates complex structures and wake patterns, especially in the super-upper branch. In this sense, modal analysis can be very useful to isolate the main structures of the system and reduce the dimensionality and complexity of the system. Among the different method for extracting flow features and

coherent structures, proper orthogonal decomposition (POD) is one of the most common techniques. POD was first introduced in fluid dynamics by Lumley [21] to decompose the randomness of turbulence into modes with some portion of the total fluctuating kinetic energy of the flow. Besides offering a new insight in the data analysis, POD in fluid dynamics is also helpful in order to create a surrogate model of the case [22].

In fact, POD has been widely used to analyse the wake of fixed cylinders [23–26]. Reduced order models in cylinders undergoing vortex-induced vibrations have been used for both finding linearized models used for prediction [27,28] and for detecting coherent structures, as presented in the present work. For the case of oscillating cylinders, Liberge and Hamdouni [29] performed the POD of a one-degree-of-freedom system at $Re_D = 1690$ using unsteady Reynolds-averaged Navier–Stokes simulations to achieve a reduced order model for the fluid structure interaction that could be used at higher Reynolds numbers.

Huera-Huarte and Vernet [30] used a method combining POD and fuzzy clustering to identify the vortex modes in digital particle velocimetry data of a circular cylinder under vortex-induced vibrations in a range of $Re_D = 1200$ to $Re_D = 12,000$ and a set of reduced velocities up to $U^* = 15$. Similarly, Riches et al. [31] studied the modes at the initial and upper branches of a 1 DoF system after extracting the snapshots using particle image velocimetry techniques. Recently, O'Neill et al. [32] presented the analysis of POD in the de-synchronisation region at $Re_D = 4000$ for a 1 DoF oscillating cylinder. They showed that the coherent motions persist in that region, thus relating the vortex-shedding frequency and the lock-in oscillation frequency of the cylinder. Using large-eddy simulations, Janocha et al. [33] studied a 1DoF cylinder at $Re_D = 3900$ and reduced velocities of $U^* = [3, 5, 7]$. With the aid of POD, they captured the dominant flow characteristics and the super harmonics associated with the vortex-shedding frequency, as well as the low-frequency modulation of the wake flow.

Some other works regarding ROM in VIV cylinders have been carried out using the variant spectral proper orthogonal decomposition (SPOD). Schubert et al. [34] presented an improved method to gain insight into the flow dynamics from a data-driven reduced-order model. They used the SPOD to reduce the data obtained from a PIV of a 1 DoF oscillating cylinder at $Re_D = 4000$. The ROM in VIV cylinders has also been applied to study other systems, as in the case of the work conducted by Zhang and Zheng [35]. They studied the non-linear dynamic states of the flow field of a two-tandem cylinder system with the downstream cylinder oscillating transversely at a low Reynolds number of $Re_D = 100$. Moreover, Mella et al. [36] used POD to analyse the span-wise wake dynamics and structural response of a bottom-fixed cylinder realizing that the free end of the cylinder had a clockwise elliptical-type trajectory.

So far, all the studies where POD has been performed in the vibrating cylinder have been carried out in 1 DoF systems, but as far the authors' knowledge is concerned, no studies have been carried out in 2 DoF systems. In the present work, POD is applied to the database generated by Pastrana et al. [16] at a $Re_D = 5300$ and reduced velocities of $U^* = 3$ and $U^* = 5.5$. The former belongs to the initial branch, while the latter is in the super-upper branch. The main objective of the present study is to shed light on the flow dynamics of the wake and to understand the interaction between the cylinder movement and the formation of the different structures in both the initial and super-upper branches. This is the first time a modal analysis of the flow in a 2 DoF oscillating cylinder is performed, which has led to the improved identification of a coherent structures, and their motion and interaction in the wake of the cylinder. In particular, in the super-upper branch, the interplay between the so-called Taylor–Görtler structures and the formation of the 2T pattern has been analysed, and the effect of this structures in the lift coefficient has been elucidated. To understand the mechanism that lead to the amplification of the cylinder movement in both the in-line and cross-stream directions are of key importance for applications such as energy harvesting [6] or to prevent structural damage due to vibrations in deep-water risers and pipelines or tall buildings.

The reminder of this manuscript is organised as follows. In the next section, a brief description of the case and the numerical method used to obtain the database used is given. Moreover, the POD technique is introduced and explained. The discussion and analysis of the main POD modes observed in both the initial branch and the super-upper branch are presented in Section 3. Finally, the conclusions are summarised in Section 4.

## 2. Material and Methods

### 2.1. Definition of the Case

The system under study is presented in Figure 2. It consists of a circular cylinder of diameter $D$, mounted with linear springs that allow the motion in the in-line and cross-flow directions. This allows the system to oscillate with two degrees of freedom. Upstream of the cylinder, the flow enters the domain with a constant velocity $U_\infty$. The flow dynamics are governed by the incompressible Navier–Stokes equations in an arbitrary Lagrangian–Eulerian (ALE) framework.

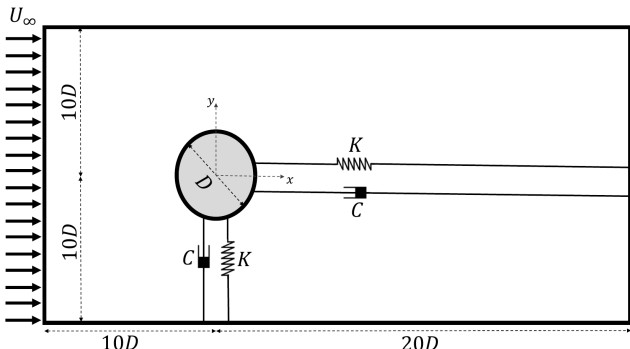

**Figure 2.** Computation domain and problem scheme.

In the present work, the database generated in Pastrana et al. [16] is used. The database is obtained at a $Re = U_\infty D / \nu = 5300$ with a low mass ratio $m^* = 2.6$ and zero damping $\xi = 0$. Here, the mass ratio $m^* = 4m/(\rho_f \pi D^2)$ is the fluid–structure mass ratio ($\rho_f$ is the fluid density and $m$ is the mass of the cylinder) and $\xi = c/(2\sqrt{k\,m})$ is the damping ratio between the structural and the critical damping. The structural stiffness and the damping are the same in both in-line and cross-flow directions, i.e., $k_x = k_y = k$ and $c_x = c_y = c$. The present study comprises the values of the reduced velocity $U^* = U_\infty/(f_n D) = 3.0$ and 5.5, which covers both the initial branch and the super-upper branch; $f_n = (2\pi)^{-1}\sqrt{k/m}$ is the natural frequency of the cylinder.

### 2.2. Summary of the Numerical Implementation

Although the numerical methodology used for generating the database was explained in Pastrana et al. [16], for completeness, a brief summary is given. Numerical results were obtained using large-eddy simulations (LESs) of the flow by means of a low-dissipation finite element (FE) scheme [37] implemented in the code Alya. Alya is a multi-physics parallel code developed for solving complex fluid mechanics problems. The low-dissipation methodology used in Alya preserves both linear and angular momentum, together with kinetic energy at a discrete level, and has been proven to yield accurate results for solving the massive separated flow past bluff bodies (see for instance two recent cases where this methodology has been successfully used [38,39]).

A non-incremental fractional-step method is used to stabilise the pressure. This allows for the use of finite element pairs that do not satisfy the inf-sup conditions, such as the equal-order interpolation for the velocity and pressure. The set of equations is integrated in time using an energy-conserving fourth-order Runge–Kutta explicit method [40] combined with an eigenvalue-based time step estimator [41]. As for the LES, the Vreman [42] sub-grid scale (SGS) model is used. To solve the fluid–structure interaction (FSI) system, an arbitrary Lagrangian–Eulerian (ALE) formulation is used; the forces at the wet surface are computed

and used to solve the second-order rigid body equations (for more details, the reader is referred to [43]).

### 2.3. Proper Orthogonal Decomposition (POD)

Proper orthogonal decomposition is a technique widely used to reduce the complexity of large random data sets and divide them into a set of deterministic functions (known as POD modes) with the objective of providing a clearer idea of the organization of the data. It does so by characterising the dominant features of the system and it is the most efficient way to capture an infinite-dimensional process with a reduced number of modes [44].

Consider a field $F(X, t)$ where $X$ represents the spatial coordinates and $t$ is the time. It can be decomposed in:

$$F(X, t) = \sum_{i=1}^{i=N} a_i(t) \Phi_i(X) \tag{1}$$

The definitions of the time coefficients $a_i(t)$ and the spatial modes $\Phi(X)$ are not unique; however, one definition of the spatial modes gives a sole result of the time coefficients. The idea of POD is finding the most suitable basis for the decomposition.

The first criterion to look at in a method to compute POD is that the basis for the spatial mode must be orthonormal such as:

$$\int_X \Phi_{i_1}(X) \Phi_{i_2}(X) \mathrm{d}x = \begin{cases} 1 \text{ if } i_1 = i_2 \\ 0 \text{ otherwise} \end{cases} \tag{2}$$

If the modes are orthonormal between each other, then the time coefficients are computed as:

$$a_i(t) = \int_X F(X, t) \Phi_i(x) \mathrm{d}x \tag{3}$$

Hence, each time coefficient only depends on its spatial mode.

All in all, it is possible to say that POD is based on finding a set of orthonormal basis vector in a subspace $R^n$ where a random vector takes its values [45]. Each vector is one POD mode and must represent a feature of the data set. All vectors have to be ordered according to its relevance in the data and the first $r$ vectors should be able to reconstruct the samples used to obtain the basis. To fulfil this, the vectors chosen to build the basis are the ones that minimise the error between the initial data set and the reconstruction performed using the POD modes.

POD can be used to treat data from a finite space by saving the variables in the domain points along several timesteps. The points do not need to have any kind of space correlation as long as they keep the same numbering along all the snapshots. The data must be ordered in a snapshot matrix $Y$ where each $M$ row represents one point of the domain and each $N$ column represents one snapshot.

In order to find the orthonormal $R^n$ basis vector from the $Y$ snapshot matrix, the single-value decomposition (SVD) is used. Similarly to the eigen decomposition, the SVD decomposes the initial matrix in three different ones:

$$Y = USV^T \tag{4}$$

Here, $U$ is a matrix of the same size as $Y$ ($M \times N$), in which the columns contain the singular vectors in each point of the domain, i.e., $U$ is the spatial modes matrix and each column represents one mode. $S$ is a $N \times N$ diagonal matrix containing the singular values of each mode in descendent order. The higher the singular value, the more energy contained in the mode; $V$ is a square matrix of size $N$, which is directly linked to the POD time coefficients. As the SVD gives the transposed right singular vectors, each row of $V$ is the time coefficient of the mode.

Let us consider a matrix $\Phi = US$ of size $M \times N$, which represents the spatial mode multiplied by its energy. As it does not depend on time, it is possible to consider this $\Phi$

as equivalent to the one in Equation (1). By grouping $U$ and $S$ together, it is possible to consider $Y = \Phi V^T$ as a decomposition that already satisfies the idea of separating the space and time dependence. The former matrix product can be expressed as the sum of $N$ matrix products:

$$Y = \sum_{i=1}^{i=N} \phi_i v_i^T \tag{5}$$

where Y is the data of the studied field $F(X, t)$, and $\phi_i$ and $v_i^T$ are the $i^{th}$ columns of $\Phi$ and $V$, respectively. As Equation (5) is equivalent to Equation (1), it can be concluded that the SVD is a decomposition that meets all the requirements to compute the POD. However, in order to reconstruct the initial database from the POD modes, it is enough to compute the matrix product from Equation (4) without any of the treatment performed to prove this equivalence. In Algorithm 1, the POD algorithm is summarised.

The POD is implemented with an in-house tool called pyLOM. PyLOM is based on compiled functions (coded with C and Cython) that can be called in *Python* scripts. The algorithm works in parallel and, in particular, the SVD has been parallelised using a binary tree algorithm, as in Demmel et al. [46]. The resulting parallel tool has a high scalability so it can handle big amounts of data, as can be seen in Figure 3. Notice that in up to 4000 processors, the scalability is nearly the theoretical one. In summary, this tool has all the needed functions to: (i) extract the fluid flow snapshots from the numerical simulations and build the snapshot matrix, $Y$; (ii) compute the single-value decomposition to extract the POD modes, as well as its energy and temporal coefficients; and (iii) post-process and visualise the results from the modal analysis. Last but not the least, it should be pointed out that for the kind of analysis to be performed here, the spatial convergence of the data is not a problem, as the data used are obtained from LES, where the mesh requirement is more restrictive than that of the POD.

---

**Algorithm 1:** Proper orthogonal decomposition.

---

> **Input:** $\mathcal{D}_i$;           // Data matrix dispersed on each processor
>
> **Output:** $U_i, S, V$;    // POD modes dispersed on each processor, singular
>               values and right singular vectors (not dispersed)
>
> /* Possibility to remove the temporal mean to compute POD of only
>      the fluctuations                                                      */

1   **if** *removeMean* **then**

2      $\mathcal{D}_{mean} = \frac{1}{n_t} \sum_{i_t=0}^{i_t=n_t} \mathcal{D}_i[:, i_t]$

3      $Y_i = \mathcal{D}_i - \mathcal{D}_{mean}$

4   **end**

5   **else**

6      $Y_i = \mathcal{D}_i$

7   **end**

8   $U_i, S, V = $ `tsqr_svd`$(Y_i)$;      // Binary tree reduction algorithm for SVD
    computation

---

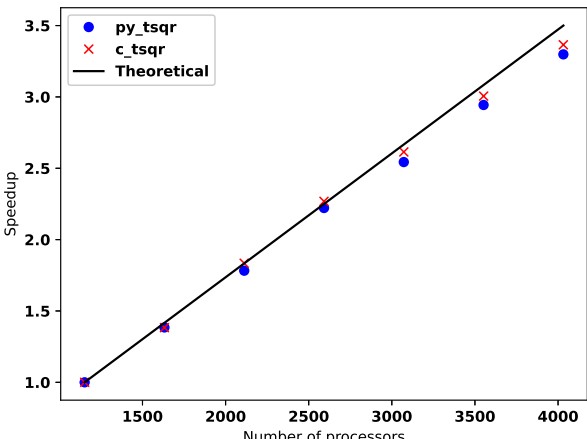

**Figure 3.** Scalability of the single-value decomposition algorithm (most computationally demanding part of POD) for the Python and Cython versions.

## 3. Results

As commented in Section 2.1, the database generated in Pastrana et al. [16] is used for POD analysis. Two different topological cases are studied, i.e., the initial branch with $U^* = 3$ and the super-upper branch with $U^* = 5.5$. In Table 1, the number of snapshots used for POD analysis and the spacing between them is given. Notice that the snapshot sampling is given in terms of the vortex-shedding frequency $St_{vs} = f_{vs}U_\infty/D$. For instance, for the case of $U^* = 3.0$, every vortex-shedding cycle is divided into 30 snapshots.

**Table 1.** The number of snapshots $N$, vortex-shedding frequency, and sampling frequency $\Delta St$ used in the POD for the different cases.

| $U^*$ | $N$ | $St_{vs}$ | $\Delta St_{vs}$ |
|-------|-----|-----------|------------------|
| 3.0 | 880 | 0.165 | 0.0054 |
| 5.5 | 944 | 0.139 | 0.0036 |

### 3.1. Flow Field Salient Features

Although described in detail in Pastrana et al. [16,20], for completeness, a brief summary of the main flow features observed in both the initial and super-upper branches is given. Figure 4 shows the coherent structures and the vortex tubes in the wake of the cylinder in both branches. As commented in the introduction, the initial branch is characterised by a 2S wake pattern, as can be observed in Figure 4a, where vortex tubes are aligned with the wake centreline. For the specific case of $U^* = 3.0$, the vortex-shedding frequency is $St = f_{vs}D/U_\infty = 0.165$ and the frequency of the cross-flow oscillations is also equal to the vortex-shedding frequency (see values summarised in Table 2). Notice also that due to the displacement of the cylinder, the drag coefficient increases by a 70% compared to the static cylinder (see, for instance, values reported by Norberg [47], Aljure et al. [48]).

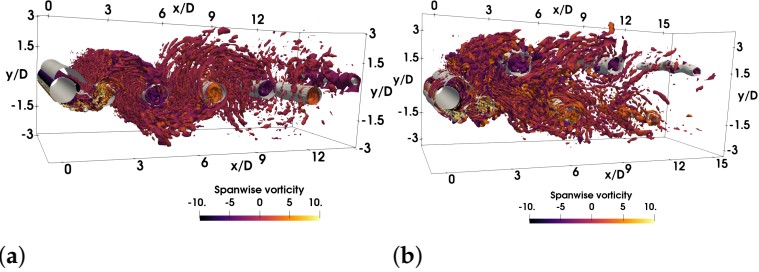

**(a)**                                                    **(b)**

**Figure 4.** Wake topology. Instantaneous Q-isocontours $Q = 3$ coloured with the span-wise vorticity together with pressure isocontours at $p = -0.5$ for (**a**) the initial branch $U^* = 3$ and (**b**) the super-upper branch $U^* = 5.5$.

The wake topology of the super-upper branch with $U^* = 5.5$ at the instant where the 2T structure is formed is depicted in Figure 4b. Compared to the initial branch, the wake is much wider and vortices shed travel downstream off the wake centreline. In the super-upper branch, the cylinder reaches the largest amplitudes in both in-line and cross-flow directions, as can be seen in Table 2. Moreover, as there is a large wake deficit as a consequence of the large displacement of the cylinder , the drag coefficient is also more than three times the values reported for the static cylinder.

**Table 2.** Summary of the main statistical flow parameters. $C_d$ is the drag coefficient; $St_{vs}$ is the vortex-shedding frequency; $f_y D/U_\infty$ and $f_x D/U_\infty$ are non-dimensional cross-flow and inline oscillation frequency of the cylinder, respectively; and $A_y/D$ and $A_x/D$ are the maximum cross-flow and in-line cylinder amplitudes, respectively.

| $U^*$ | $C_d$ | $St_{vs}$ | $f_y D/U_\infty$ | $f_x D/U_\infty$ | $A_y/D$ | $A_x/D$ |
|-------|-------|-----------|------------------|------------------|---------|---------|
| 3.0 | 1.709 | 0.165 | 0.166 | 0.331 | 0.186 | 0.091 |
| 5.5 | 3.235 | 0.141 | 0.141 | 0.282 | 1.200 | 0.322 |

In addition to the 2T vortex-shedding pattern, the super-upper branch is characterised by the intermittent emergence of three-dimensional structures of counterrotating stream-wise vorticity on the cylinder surface [20]. These structures (see Figure 5) have been shown to be due to the appearance of a centrifugal instability when the cylinder approaches its maximum elongation.

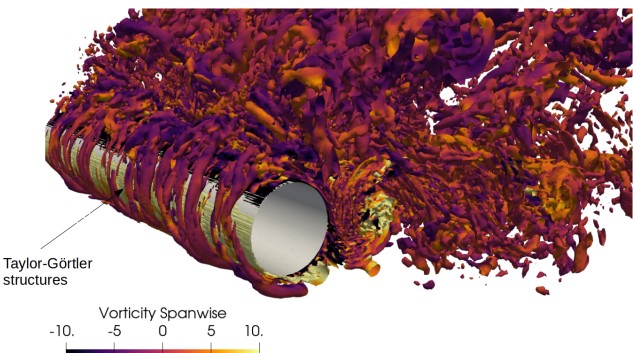

**Figure 5.** Taylor-Görler structures identified by iso-surfaces of the non-dimensional Q-criterion $Q = 3$ coloured with the span-wise vorticity for the case of $U^* = 5.5$.

*3.2. Initial Branch, $U^* = 3$*

Figure 6a shows the percentage of the turbulent kinetic energy (*tke*) captured by each of the POD modes and Figure 6b shows the cumulative energy for all the modes. The POD method ranks the modes according to their energy content in descending order. Notice that in the figure, only the first 20 modes are plotted. As higher modes are expected to have lower *tke* and a more disorganised structure, these modes contribute little to the present discussion and are not plotted. The two highest energy containing modes account for roughly 44% and 32% of the flow *tke*. These two modes contain the main coherent structures of the wake, i.e., the span-wise vortex tubes. However, for high-order modes, the energy is dispersed. In fact, as can be seen from Figure 6b, 57 modes are required to capture 90% of the *tke* and more than 230 to capture the 95%. This is in contrast with POD analysis performed on the static cylinder, where most of the energy is captured by a few modes. For instance, Ma et al. [23] at $Re = 3900$ found that most of the energy is captured by the first 20 modes. The poorer energy convergence observed here can be attributed to a re-distribution of the turbulent kinetic energy to high-order modes due to the cylinder oscillation. Considering that in the present work, there is a combination of in-line and cross-stream oscillation of the cylinder, a larger energy spread is expected. Although, *a-priori*,

this poor energy convergence might render the construction of a reduced-order model containing 95% of the energy impossible, it is still useful in order to study the main energy containing modes and to separate the largest scales of the flow to study their dynamics.

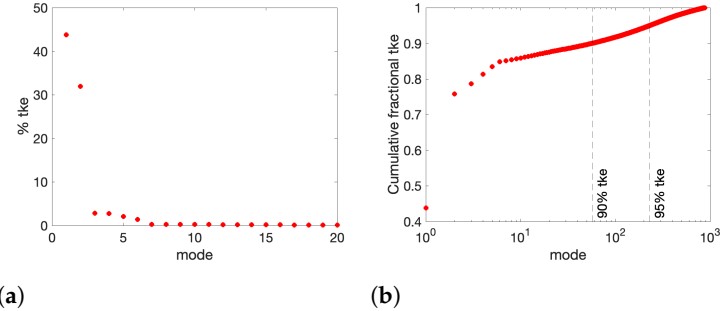

**(a)**　　　　　　　　　　　　　　**(b)**

**Figure 6.** Initial branch $U^* = 3.0$. (**a**)Energy of the first 20 POD modes expressed as a percentage of the turbulent kinetic energy (*tke*), (**b**) cumulative energy.

The first six spatial modes of the stream-wise and cross-stream velocity fluctuations are presented in Figure 7. They are plotted at the mid-span of the periodic direction z. As the vortex shedding is a periodic phenomenon, it is expected to obtain a pair of highly correlated modes (i.e., 1–2, 3–4, ...) with a phase shift of 90° between each other. This phase shift can be readily seen if the distances between the centres of the same correlated areas are measured. In fact, if the lissajous of the time coefficients for these modes is plotted (see Figure 8), one can see the circular representation of the pair of coefficients; this suggests a cyclic variation between them. This distribution is expected as large-scale vortexes are formed with a periodic pattern and convected downstream forming the 2S wake pattern.

The first pair of modes contributes to the formation of the main wake pattern, i.e., they represent the main vortex tubes typical of a von Kármán vortex street. This can be readily seen from the spatial modes (Figure 7a–d, where contours with positive and negative values can be interpreted as an alternating vortex structure. Moreover, the main vortexes can also be seen from the superposed velocity vectors obtained from the contribution of both velocity components. The convective downstream motion of these vortexes is apparent by their phase shift, as commented before. As their contribution is directly related with the large-scale vortex pattern of the wake, the energy spectrum of the coefficients for these first two modes (Figure 9a) peaks at the vortex-shedding frequency, i.e., at $St_{vs} = 0.165$. This also coincides with the frequency of the cross-stream displacement of the cylinder (see values summarised in Table 2). Moreover, the spectrum also shows a strong peak at $St = 0.494$, which corresponds to the third harmonic of the vortex-shedding frequency. The presence of high-order harmonics of the vortex-shedding frequency can be attributed to an increase in the velocity fluctuations and the non-linear nature of the flow [49] due to the vortex-shedding process and the wake pattern evolution.

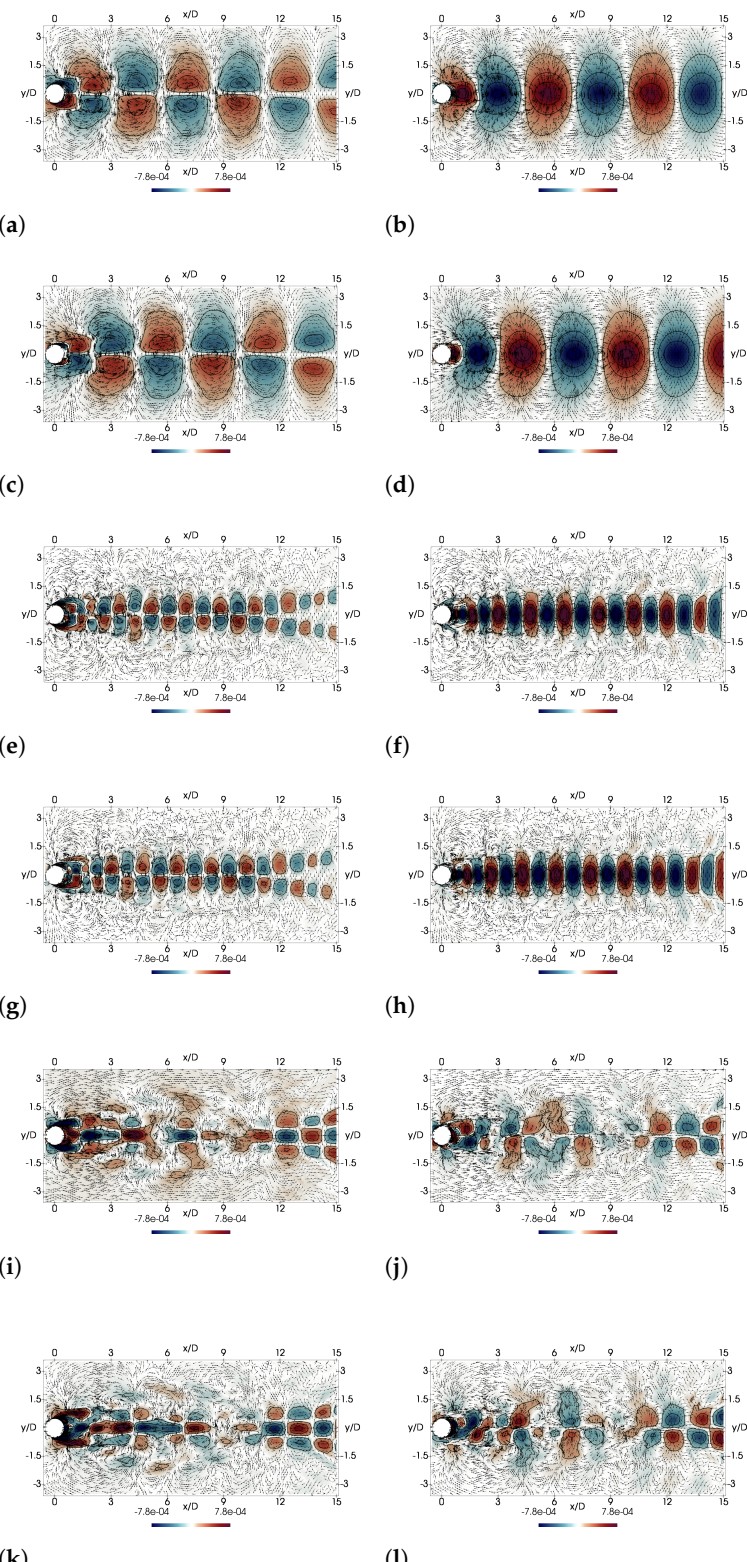

**Figure 7.** First six modes of the stream-wise (**a**,**c**,**e**,**g**,**i**,**k**) and cross-stream velocity fluctuations (**b**,**d**,**f**,**h**,**j**,**l**) in the initial branch.

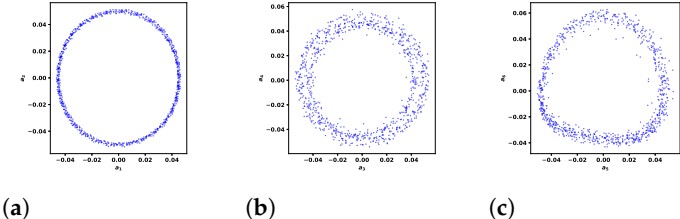

(**a**)    (**b**)    (**c**)

**Figure 8.** Coefficients phase diagram. (**a**) Modes 1 and 2; (**b**) modes 3 and 4; (**c**) modes 5 and 6.

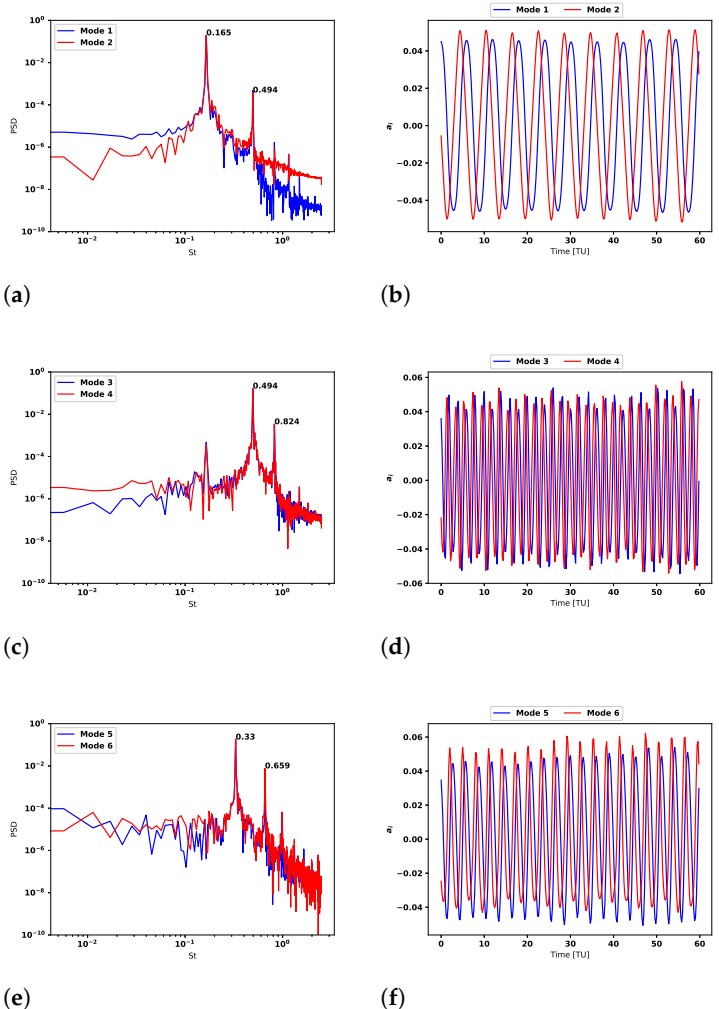

(**a**)    (**b**)

(**c**)    (**d**)

(**e**)    (**f**)

**Figure 9.** Energy spectra (**a**,**c**,**e**) and the first 60 TU of the time coefficients (**b**,**d**,**f**) of each of the three pairs of modes: 1–2, 3–4, and 5–6, respectively.

Although the vortex-shedding frequency and main vortex tubes are predicted by the first two modes, in order to capture the cylinder movement and convective motion of the wake, higher modes should be considered. In fact, modes 3 and 4 can be seen as a first correction to the main wake vortexes. The correlations found in these two modes (Figure 7e–h) enhance the center of the vortexes predicted by the first two modes, while they reduce their diameter. These modes have two dominant frequencies, at $St = 0.494$ and $St = 0.824$, which are the third and fifth harmonics of the vortex-shedding frequency detected in the first and second modes. It can be argued, then, that these modes can be associated with the structures responsible of higher-velocity fluctuations. If these two modes are also considered, it would be possible to partly recover the downstream convective movement of the vortexes in the wake, though some additional structures and dynamics might not be well reproduced.

Modes 5 and 6 are presented in Figure 7i–l. The observed correlations are far more chaotic than those from previous modes; this indicates that they are linked with smaller scales of the flow. Another observation is that the centre of the detected structures is located in the stream-wise axis, which might suggest that this mode contains information about the structures and movement of the vortices linked to the in-line vibration of the cylinder. In fact, the energy spectrum of these two modes, presented in Figure 9, peaks at $St = 0.33$, i.e., the frequency of the oscillations of the cylinder in the stream-wise direction (see values summarised in Table 2). It also has another peak at $St = 0.659$, which corresponds to the second harmonic of said frequency.

In Figure 10, the span-wise vorticity isocontours of the reconstructed flow with 2, 10, and all the modes, respectively, is depicted. Although the reconstructed flow with only two modes exhibits the location of the main wake vortex tubes (Figure 10a), a better representation can be obtained using the ten most energetic modes (Figure 10b). This reconstruction, which contains the information of the generation and dissipation of the main structures discussed in this section, is a rather good representation of the larger scales of the flow (see also the original flow with all the structures of the turbulent wake in Figure 10c). In fact, it can be seen as an ensemble averaged of the flow (see for instance Figure 7.4 in Rodríguez et al. [50], where the ensemble averaged flow for this case is depicted), where small-scale structures have been filtered out, but still the wake retains the information of the large coherent structures.

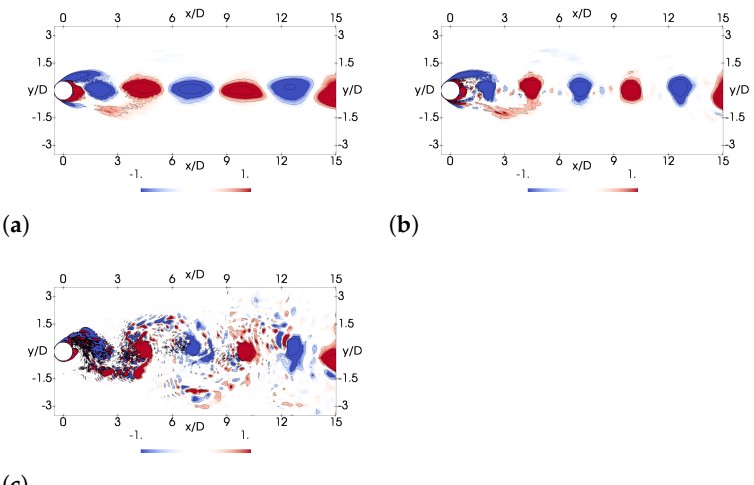

(a)

(b)

(c)

**Figure 10.** Instantaneous vorticity isocontours for a flow reconstructed with (**a**) 2 modes; (**b**) 10 modes; and (**c**) all modes.

### 3.3. Super-Upper Branch

The turbulent kinetic energy of the first 20 POD modes and the cumulative *tke* are given in Figure 11. As discussed in the initial branch, energy is re-distributed among high-order modes. In this particular case, the energy of the first and second modes account for 30% and 17% of the total kinetic energy, but 335 modes are required to capture 90% of the energy and 539 for the 95%. As previously remarked, the re-distribution of the energy is associated with the cylinder displacement in both in-line and cross-stream directions. In the super-upper branch, the cylinder reaches its larger amplitudes in both directions; thus, a larger spreading of energy can be expected over higher-order modes, as can be seen in the figure. Nonetheless, as in the initial branch, as the first modes contain a large part of the energy, representing the large-scale structures, it is still possible to use POD for studying these coherent structures, their main frequencies, and the impact on cylinder forces.

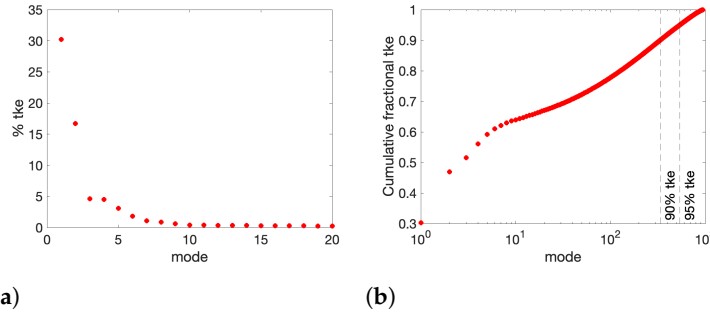

**(a)**                                                    **(b)**

**Figure 11.** Super-upper branch $U^* = 5.5$. (**a**) Energy of the first 20 POD modes expressed as a percentage of the turbulent kinetic energy (*tke*) and (**b**) cumulative energy.

Figure 12 shows the first six spatial modes of the stream-wise and cross-stream velocity components and Figure 13 depicts the time coefficients and energy spectra of these modes. As for the case of the initial branch, the spatial modes are presented in the mid-plane of the span-wise direction. The first couple of stream-wise and cross-stream velocity spatial modes are represented in Figure 12a–d. Modes 1 and 2 of the stream-wise velocity show four different correlated areas. By analysing Figure 12a,c, an antisymmetric pattern is seen. From the top to the symmetry plane, it is possible to find three blue spots followed by a large red region. As the pattern is antisymmetric, their corresponding ones can be seen in the bottom side. This means that in the wake, there are six large-scale vortexes. The cross-stream velocity modes (Figure 12b,d) corroborate this distribution. Notice that one of the main differences with the initial branch is that vortexes travel downstream off the centre, parallel to the wake centreline. The energy spectra of these two modes (see Figure 13a) exhibit a high-energy peak at the vortex-shedding frequency $St_{vs} = 0.141$. Moreover, two additional peaks are observed. Mode 1 has its second peak at $St = 0.418$ and mode 2 at $St = 0.701$; these are the third and fifth harmonics of the vortex-shedding frequency, respectively. The presence of these harmonics is the footprint of the 2T vortex-shedding pattern. In fact, the time coefficient signal resulting from the POD analysis is different to the one obtained in the 2S vortex-shedding mode. If the signal of these coefficients is inspected, one can see a change in the slope just after the signal reaches its maximum (minimum) value. In this case, although there is a 90° phase shift between these two modes, the formation and shed of the vortexes that form the 2T pattern leave a footprint in these coefficients. As a consequence, the phase diagram is not a perfect circle; instead, a square with rounded corners is required, as can be seen in Figure 14.

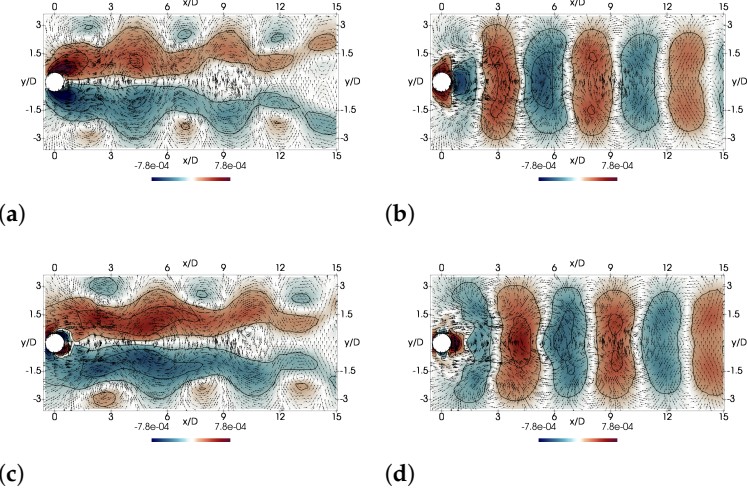

**(a)**                                                    **(b)**

**(c)**                                                    **(d)**

**Figure 12.** *Cont.*

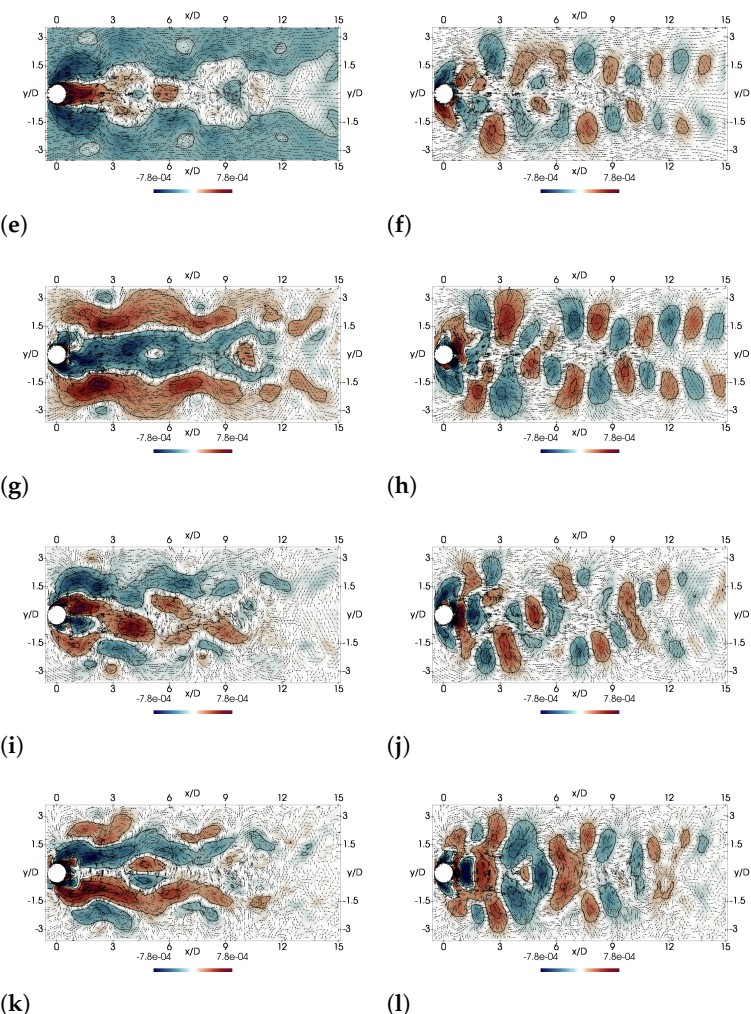

**Figure 12.** First six modes of the stream-wise (**a,c,e,g,i,k**) and cross-stream velocity (**b,d,f,h,j,l**) at the super-upper branch.

Figure 12e–h show the third and fourth modes of the stream-wise and cross-stream velocities. As explained in the initial branch, information from these two modes is complementary to understand the vortex's convection in the wake. Moreover, when these modes are analysed, a clear change in the pattern around $x/D = 7$ can be seen. As will be seen in the reconstructed flow, this first zone between the cylinder and $x/D = 7$ corresponds to the vortex formation zone, i.e., comprising the zone where vortexes are formed and shed into the wake. Downstream $x/D = 7$, a more regular pattern is observed; this corresponds to the formed wake where vortexes travel parallel to the wake centreline.

Associated with these modes is the frequency of the in-line oscillation of the cylinder, i.e., $St = 0.282$, which can also be interpreted as an harmonic of the vortex-shedding frequency. This frequency can be identified as a strong peak in the spectra of the coefficients of modes 3 and 4 (see Figure 13c). Similarly to the first couple of modes, there are secondary peaks in the spectra; mode 4 has an additional peak at the third harmonic of the vortex-shedding frequency, $St = 0.418$. Indeed, the vortex-shedding process in this branch is more complex as it implies the formation of the 2T structure and the Taylor–Görtler structures [20]. As will be discussed below, the formations of both structures are related but occur at different instants of the cycle; this leaves a different footprint in the harmonics observed in this couple of modes. In fact, these modes are the first to contain information on the Taylor–Görtler structures. This information can be found in the span-wise velocity component correlations. Figure 15 shows the third and fourth modes of the velocity along

the span-wise direction. If these modes are examined, one can see the information related to three-dimensional structures and specifically those of the Taylor–Görtler structures. However, it has to be pointed out that similar correlations are also seen in other less energetic modes, which means that to complete the formation and dissipation of these structures, the inclusion of more modes is necessary.

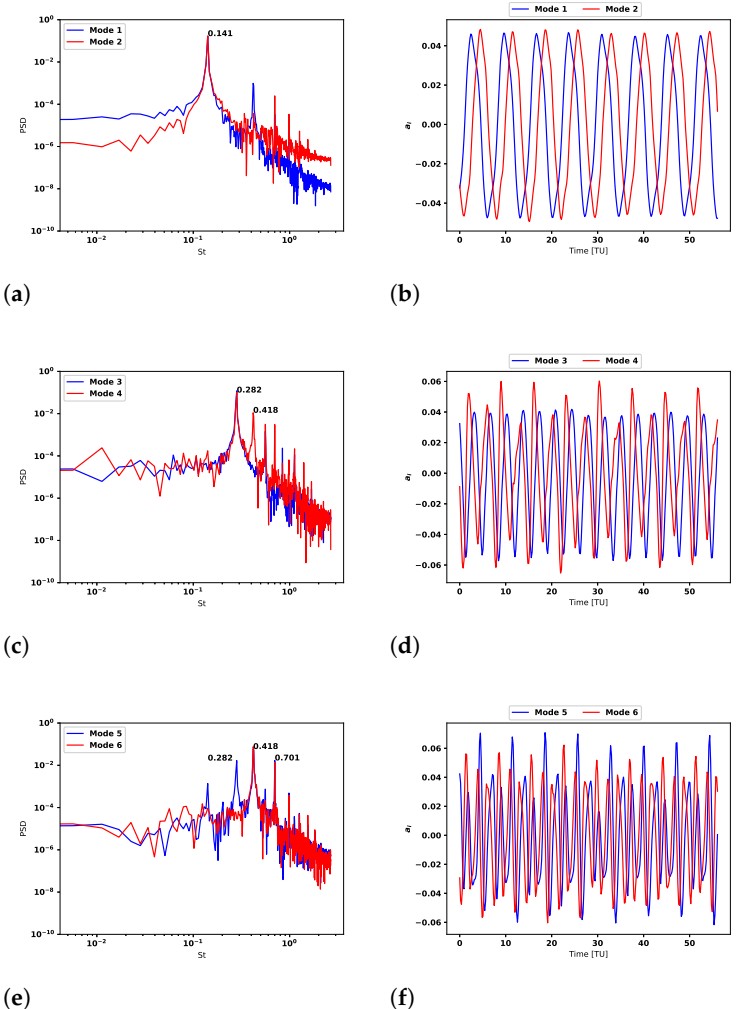

**Figure 13.** Energy spectra (**a**,**c**,**e**) and the first 60 TU of the time coefficients (**b**,**d**,**f**) of each of the three pairs of modes: 1–2, 3–4, and 5–6, respectively.

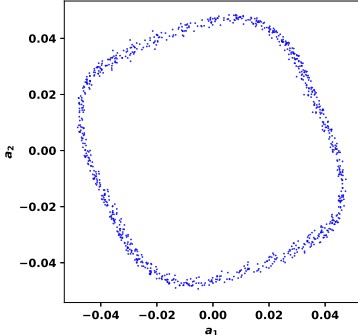

**Figure 14.** Phase diagram of the first two modes in the supper-upper branch.

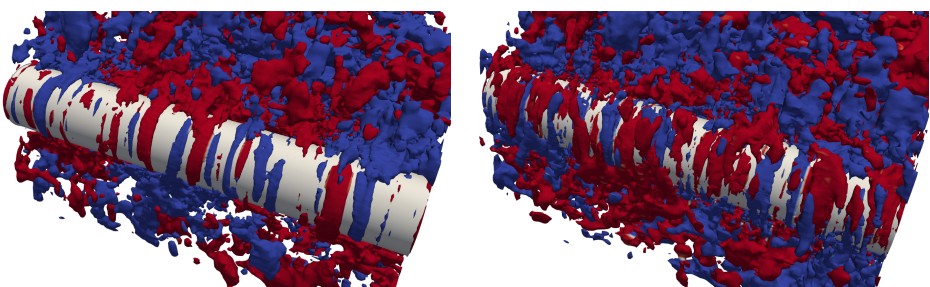

**Figure 15.** Third and fourth mode span-wise velocity contours ($\pm 2.5 \times 10^{-5}$).

Modes 5 and 6 of the stream-wise and cross-stream velocities (see from Figure 12i–l) are the first that include the formation of the 2T wake structure. The appearance of the 2T can be directly read from the stream-wise correlations as they have several extreme changes next to the cylinder, which imply the presence of the vortex rotating in different senses. Figure 13e shows the spectra for both modes. The most energetic peak in the spectra is located at $St = 0.418$, which is actually the third harmonic of the vortex-shedding frequency. In addition to this peak, a peak at the fifth harmonic of the vortex-shedding frequency is observed, i.e., at $St = 0.701$ and mode 5 also has a secondary peak at $St = 0.282$. This is the frequency of the in-line oscillation of the cylinder.

The reconstruction of the flow in the wake with 2, 6, 10 and all the modes is shown in Figure 16, where contours of the span-wise vorticity are depicted. Even though the formation of the large-scale vortexes can already be seen with two modes, it is after the inclusion of the first six modes that the formation of the 2T pattern can be observed. This structure, marked in Figure 16b, can be seen as two counter-clockwise-rotating vortexes and one clockwise-rotating vortex close to the cylinder. Moreover, for capturing the formation and dissipation of the Taylor–Görtler structures, at least 10 modes are required, as can also be seen in Figure 17. In the figure, the formation of these structures at different points of the cycle is represented by stream-wise vorticity contours.

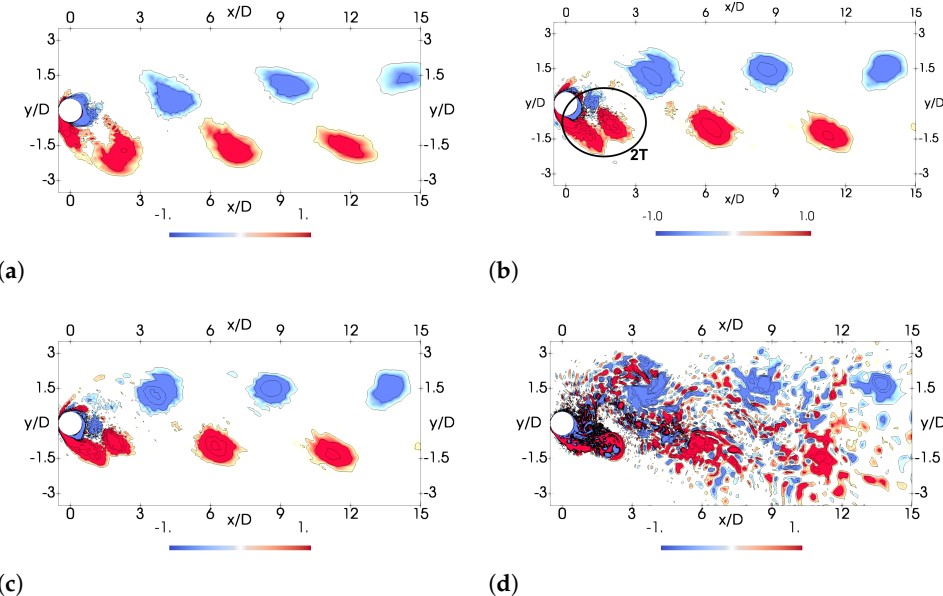

**Figure 16.** Reconstructed flow: instantaneous span-wise vorticity iso-contours with (**a**) 2 modes, (**b**) 6 modes, (**c**) 10 modes, and (**d**), all modes.

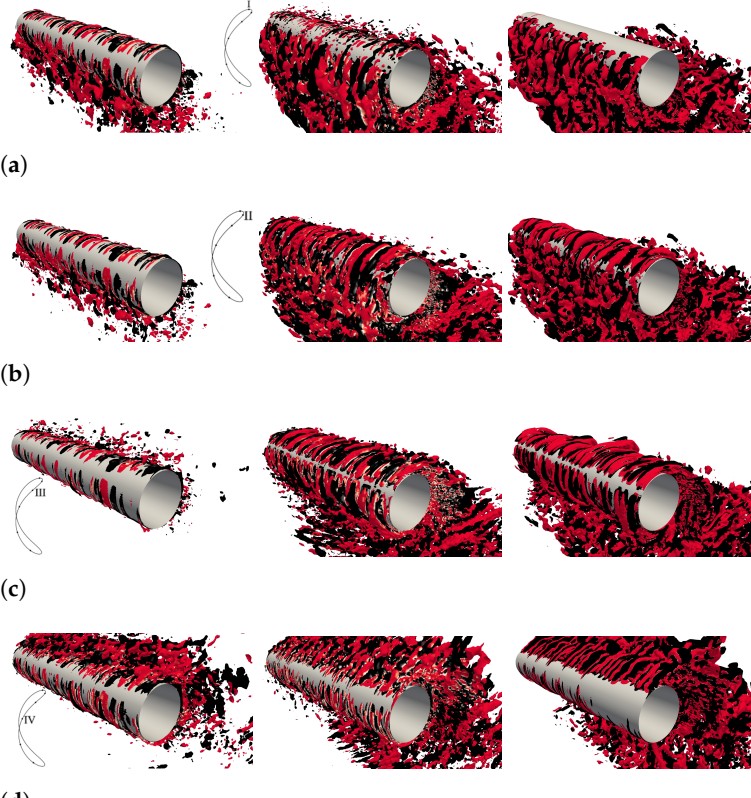

**Figure 17.** Stream-wise vorticity iso-contours of the reconstructed flow showing the Taylor–Görtler structures. From left to right: 2, 10, and all modes. Results presented at different instants of the cross-flow oscillation cycle are shown in the figure. Each image is represented every $1/8$ of the cycle starting from the maximum elongation in $y$. (**a**) $t = 0$, (**b**) $t = 1/8\tau$, (**c**) $t = 1/4\tau$, (**d**) $t = 3/8\tau$.

Although the formation of the Taylor–Görtler (TG) structures was reported by Pastrana et al. [20], a decision on whether these structures are related to the 2T pattern described by Jauvtis and Williamson [17] could not be established. In this sense, the present POD analysis can be useful to analyse if the formation and dissipation of the Taylor–Görtler (TG) structures and the 2T pattern are two phenomena that are interrelated. POD can offer additional insight into the coherent structures of the flow and provide a better understanding of the interaction between the 2T formation and the TG structures, as well as the effect of these structures in the lift force.

Figure 18 shows the evolution of the lift force, the wake pattern, and the TG structures along half-vortex-shedding cycle. The wake pattern is represented with the span-wise vorticity iso-contours of the flow reconstruction using the first 10 POD modes. Moreover, TG structures are depicted using stream-wise vorticity contours. By using only 10 POD modes, it is possible to have the large-scale motions which roughly contain about 64% of the *tke*. In this sense, the evolution of these large-scale motions can be analysed in order to understand their dynamics and the role on the cylinder forces. Figure 18b corresponds with point A in the lift curve, i.e., the instant when the cylinder reaches its maximum cross-flow elongation and the lift force reaches its maximum value. This is the instant when the counter-clockwise-rotating vortex (CCR) is shed into the wake. At this instant, a new clockwise-rotating vortex is formed in the rear end of the cylinder (identified as CR1 in the figure). In fact, CR1 forms as a consequence of the shear-layer roll-up; a low-pressure centre is formed in that location (not shown). Moreover, TG structures are formed as a consequence of the centrifugal instability and appear at the cylinder maximum elongation (see Pastrana et al. [20]). Incipient vortical structures are also started to form from ejected flow from the TG structures.

As the cylinder falls back towards the centreline, it leaves behind CR1. The separation of this low-pressure centre favours the growth of CR2 and CCR3 (see Figure 18c). The latter can be seen as a small spot in between CR1 and CR2. Notice also that at this position, TGs are almost dissipated, and apex transitional structures can be observed close to the cylinder.

A combination of the downward and backward movement when the cylinder passes by its rearmost stream-wise position promotes the engulfment of CR3 (Figure 18d). This makes it possible for CR1 and CR2 to merge (point C in the lift curve). The engulfment of CCR3 by the cylinder has two direct consequences: it gives rise to a momentary extra increase in the lift force and the formation of a new low-pressure centre in the back of the cylinder (Figure 18e). The former is relevant as it contributes to a larger cylinder elongation in the cross-stream direction. Eventually, the new low-pressure centre will become the seed for a new 2T structure. Notice that the presence of the 2T pattern is only observed in the vortex formation zone, but after the shedding process, only single vortexes are convected downstream off the wake centreline.

By analysing the formation and dissipation of these structures in detail, one can argue that TG structures are directly responsible for the formation of the 2T vortex pattern. Moreover, the interplay between TGs and the cylinder movement provokes the ejection of structures away from the cylinder and is the responsible for the vortex-shedding process. This is in contrast with what is observed in the fixed cylinder and in vibrating cylinders with 2S and 2P patterns, in which vortexes are shed by the interaction of both shear layers and the entrainment of the irrotational fluid. All in all, the present analysis contributes to a better understanding of the wake dynamics in the super-upper branch as well as to shed light on the role of the large-scale motions on the forces on the cylinder and in the amplification of the cylinder movement, especially in the cross-stream direction.

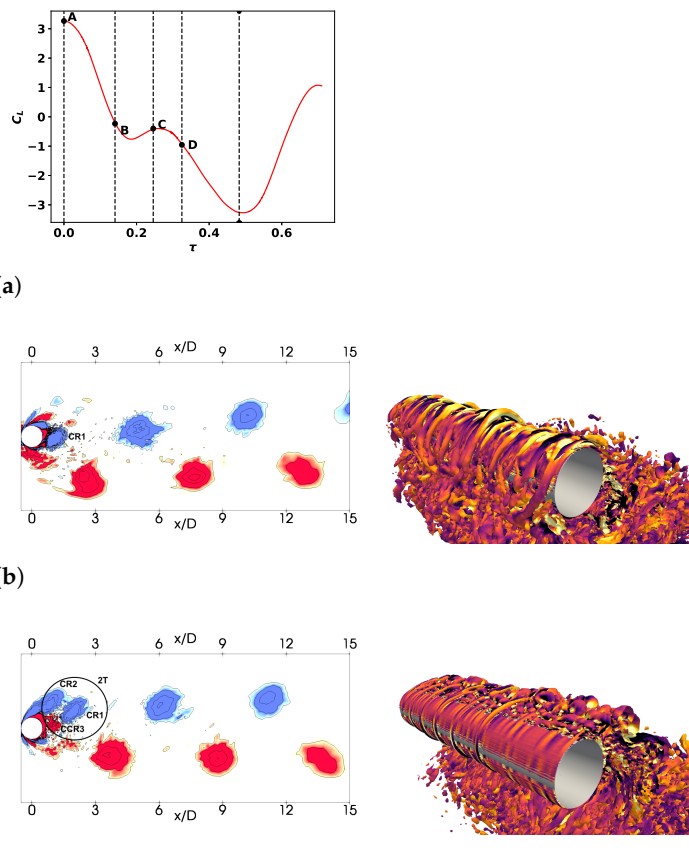

**Figure 18.** *Cont.*

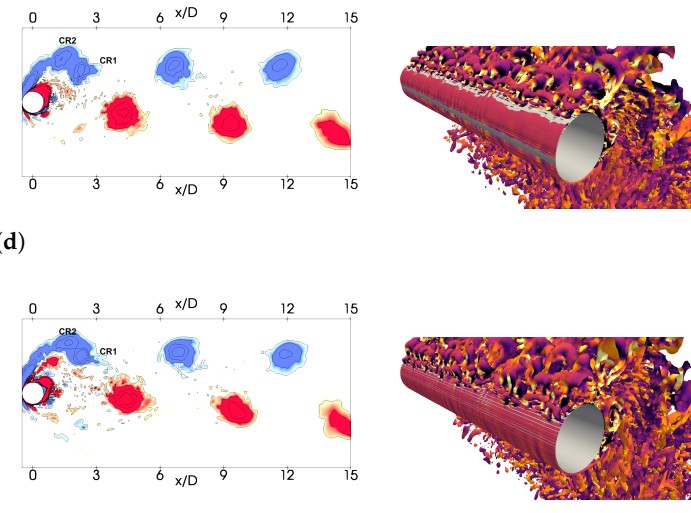

(d)

(e)

**Figure 18.** Coherent structures evolution along the half-vortex-shedding cycle of the cylinder reconstructed with 10 modes. (**a**) Lift coefficient evolution; (**b**) span-wise vorticity contours (left) and stream-wise vorticity contours at point A of the lift curve; (**c–e**) the same as at points B, C, and D, respectively. CCR and CR stand for counter-clockwise- and clock-wise-rotating vortexes, respectively.

## 4. Summary

In the present study, POD is used to investigate the wake dynamics and free vortex formation to oscillate two-degrees-of-freedom low mass ratio circular cylinder at $Re = 5300$. The database generated by Pastrana et al. [16] using large-eddy simulations is used here to analyse the flow at two reduced velocities: $U^* = 3.0$ and $U^* = 5.5$. The former corresponds with the initial branch, characterised by a 2S vortex pattern, while the latter corresponds with the super-upper branch characterised by the so-called 2T vortex pattern. This is the first time this kind of analysis is performed in a two-degree-of-freedom oscillating cylinder to gain insight into the vortex dynamics and its implication on the forces acting on the cylinder.

Around 1000 non-correlated in-time velocity fields are decomposed using the snapshot method. It is observed that the first two modes capture a large percentage of the turbulent kinetic energy but in order to capture 95% of the TKE, and to have a good representation of the flow turbulent statistics, 230 and 539 modes are required in the initial and super-upper branches, respectively. This is attributed to the cylinder in-line and cross-flow movement that redistributes the energy to high-order modes. With such energy spread, it is impossible to construct a reduced-order model of the flow. However, it has been shown that with a few modes, it is still possible to capture the dynamics of the large-scale vortex formation and wake dynamics, which are relevant to understanding the vortex-shedding process in both branches, and in particular the intricacies of the 2T vortex pattern.

The modal analysis has been useful to elucidate the main structures of the wake. Overall, the decomposed flow in the initial branch exhibits a mode distribution similar to both the one-degree-of-freedom vibrating cylinder and the static cylinder. It is found that the first couple of modes are representative of the large-scale vortical structures of the wake and the frequency associated with this pair of modes is related to the cross-flow cylinder movement and also with the vortex-shedding frequency. The spatial mode distribution observed in the super-upper branch is totally different to that obtained in the initial branch. While the first pair of modes still represents the large-scale wake vortexes, sub-harmonics detected in the coefficient's signal reveal the formation of the 2T vortex pattern. Moreover, the presence of near-wake three-dimensional structures is contained into the span-wise modes of the velocity field. Actually, Taylor–Görtler structures are seen to emerge in the

third and fourth modes, although to understand the dynamics of their formation and dissipation, reconstruction of the flow using high-order modes is necessary.

Last but not the least, the relationship between the Taylor–Görtler structures and the 2T vortex formation pattern is unravelled by analysing the flow reconstructed with the first ten modes. It is seen that 2T vortexes are formed by virtue of the ejection of the stream-wise Taylor–Görtler vortexes when the cylinder moves from its maximum elongation towards the wake centreline. Actually, it is observed that the momentary increase in the lift force half-way through both lift extrema is due to the cylinder engulfment of the third vortex that conforms the 2T structure. This is also the reason why the triplet of vortexes is only observed in the vortex formation zone, but in the wake, only single vortexes are convected downstream of the wake centreline. Moreover, as a result of this extra lift force, larger elongation in the cylinder movement can be attained, which is relevant when it comes to application, such as energy harvesting. In such applications, the maximisation of forces acting on the cylinder results in a higher-energy transfer from the flow to the system.

**Author Contributions:** Conceptualization, I.R. and O.L.; methodology, I.R and O.L.; software, B.E., O.L., and A.M.; data curation, J.C.C.; formal analysis, I.R. and B.E.; investigation, B.E., I.R., and O.L.; writing—original draft preparation, B.E. and I.R.; writing—review and editing, all authors. All authors have read and agreed to the published version of the manuscript.

**Funding:** This work has been partially financially supported by the Ministerio de Economía, Industria y Competitividad, Secretaría de Estado de Investigación, Desarrollo e Innovación, Spain (Ref. PID2020-116937RB-C21, PID2020-116937RB-C22), and by the European High-Performance Computing Joint Undertaking (JU) under grant agreement No 956104. The JU receives support from the European Union's Horizon 2020 research and innovation programme and Spain, France, Germany. O. Lehmkuhl work is financed by a Ramón y Cajal postdoctoral contract by the Ministerio de Economía y Competitividad, Secretaría de Estado de Investigación, Desarrollo e Innovación, Spain (RYC2018-025949-I)).

**Institutional Review Board Statement:** Not applicable.

**Informed Consent Statement:** Not applicable.

**Data Availability Statement:** Raw data were generated at SuperMUC supercomputer. Derived data supporting the findings of this study are available from the corresponding author I.R. on request.

**Acknowledgments:** We also acknowledge PRACE for awarding us access to SuperMUC supercomputer (Project ViValdi Ref. 2017174222).

**Conflicts of Interest:** The authors declare no conflicts of interest.

## Nomenclature

| | |
|---|---|
| $A$ | movement amplitude |
| $a(t)$ | temporal coefficients |
| $c$ | damping |
| $C_d$ | drag coefficient |
| $D$ | cylinder diameter |
| $F(X,t)$ | spatial and temporal field |
| $f$ | frequency |
| $f_n$ | natural frequency of the cylinder |
| $k$ | structural stiffness |
| $L$ | cylinder length |
| $m$ | mass of the cylinder |
| $m^* = 4m/(\rho \pi D^2 L)$ | mass ratio |
| $M$ | number of spatial points |
| $N$ | number of modes |
| $Re = U_\infty D/\nu$ | Reynolds number |
| $S$ | singular values (energy of the modes) |
| $St = fD/U_\infty$ | Strouhal number |

| | |
|---|---|
| $t$ | time |
| $U$ | left singular vectors (POD modes) |
| $U_\infty$ | freestream velocity |
| $U^* = U_\infty / f_n D$ | reduced velocity |
| $V$ | right singular vectors (temporal coefficients) |
| $v_i$ | ith singular vector |
| $X$ | spatial coordinates |
| $Y$ | snapshot matrix |
| Greeks | |
| $\rho$ | fluid density |
| $\Phi(X)$ | spatial modes |
| $\phi_i$ | ith spatial mode |
| $\xi = c / (2\sqrt{km})$ | damping ratio |
| $\nu$ | kinematic viscosity |
| Subscripts | |
| $T$ | transposed |
| $vs$ | vortex-shedding |
| $x$ | stream-wise |
| $y$ | cross-stream |
| Acronyms | |
| ALE | arbitrary Lagrangian–Eulerian |
| CCR | counter-clockwise-rotating |
| CR | clockwise-rotating |
| DoF | degrees of freedom |
| FE | finite element |
| FSI | fluid structure interaction |
| LES | large-eddy simulations |
| PIV | particle image velocimetry |
| POD | proper orthogonal decomposition |
| ROM | reduced-order model |
| SGS | sub-grid scale |
| SPOD | spectral proper orthogonal decomposition |
| SVD | single-value decomposition |
| TG | Taylor–Görtler |
| tke | turbulent kinetic energy |
| VIV | vortex-induced vibrations |

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
