# Peer review of "On the Wake Dynamics of an Oscillating Cylinder via Proper Orthogonal Decomposition"

_fluids, doi:10.3390/fluids7090292_

Round 1

Reviewer 1 Report

Please see attached PDF for comments and suggestions for authors.

Author Response

Reply in attached file

Reviewer 2 Report

The article used simulation data from reference [17] data and presented the POD used for the analysis of the vortex topology for two different cases. It must be noted that the paper is very methodologically refined and the use of the data from the simulation described in another very good journal is a big advantage. The description and the analysis of the results were made correct. The data presented in the figures and in the text complement each other to form a concise whole. The conclusion of the paper is brief and lapidary. The references were chosen correctly, although the names of the authors appear there quite often. I only have four minor comments.

-In figure 1, the signs (L, U, etc.) should be explained in the figure description. Figure 1 is disturbingly similar to fig. 1 from [17], maybe add the citation.

- In section 2.2, the kind and properties of fluid should be defined. The article [17] is not open access, so access to simulation data may be difficult.

- The nomenclature section should be added. The number of used symbols is significant and the nomenclature section will greatly simplify reading the article.

- Have persons conducting simulations from the article [17] agreed to use simulation data? I know that two authors are also responsible for the article [17], but it is quite an important issue.

I recommended this article for publication.

Round 2

Reviewer 1 Report

The authors have well-addressed the comments and suggestions. I recommend the paper for publication.